# RELVIT: CONCEPT-GUIDED VISION TRANSFORMER FOR VISUAL RELATIONAL REASONING

**Xiaojian Ma**[1] , **Weili Nie**[2] , **Zhiding Yu**[2] , **Huaizu Jiang**[3] , **Chaowei Xiao**[2,4] , **Yuke Zhu**[2,5] ,
**Song-Chun Zhu**[1] , **Anima Anandkumar**[2,6]
[1]UCLA  [2]NVIDIA  [3]Northeastern University  [4]ASU  [5]UT Austin  [6]Caltech
{xiaojian.ma@, sczhu@stat.}ucla.edu
{wnie,zhidingy,chaoweix,aanandkumar}@nvidia.com
h.jiang@northeastern.edu, yukez@cs.utexas.edu

## ABSTRACT

Reasoning about visual relationships is central to how humans interpret the visual
world. This task remains challenging for current deep learning algorithms since
it requires addressing three key technical problems jointly: 1) identifying object
entities and their properties, 2) inferring semantic relations between pairs of en-
tities, and 3) generalizing to novel object-relation combinations, *i.e.* systematic
generalization. In this work, we use vision transformers (ViTs) as our base model
for visual reasoning and make better use of concepts defined as object entities and
their relations to improve the reasoning ability of ViTs. Specifically, we intro-
duce a novel *concept-feature dictionary* to allow flexible image feature retrieval
at training time with concept keys. This dictionary enables two new concept-
guided auxiliary tasks: 1) a *global* task for promoting relational reasoning, and
2) a *local* task for facilitating semantic object-centric correspondence learning.
To examine the systematic generalization of visual reasoning models, we intro-
duce systematic splits for the standard HICO and GQA benchmarks. We show the
resulting model, *Concept-guided Vision Transformer* (or RelViT for short) signif-
icantly outperforms prior approaches on HICO and GQA by 16% and 13% in the
original split, and by 43% and 18% in the systematic split. Our ablation analyses
also reveal our model's compatibility with multiple ViT variants and robustness to
hyper-parameters.

## 1 INTRODUCTION

Deep neural networks have achieved great success in visual recognition. However, their ability for
visual relational reasoning, *i.e.* reasoning with entities and their relationships in a visual scene,
still falls short of human-level performances, especially in real-world domains. The challenges of
common visual relational reasoning tasks, *e.g.* HICO and GQA benchmarks (Chao et al., 2015;
Hudson & Manning, 2019) are manifested in three aspects: 1) **object-centric learning** to identify
objects (including humans) as well as their visual properties; 2) **relational reasoning** to infer all
pairwise relationships between the object entities; and 3) **systematic generalization** to reason with
visual entities and relations on novel object-relation combinations and extrapolate to longer rea-
soning hops (Bahdanau et al., 2018; Hupkes et al., 2020). While existing models have leveraged
pre-trained object detectors (Ren et al., 2015; Jiang et al., 2020) and/or explicit symbolic reasoning
methods (Yi et al., 2018) to tackle these challenges, they leave ample space for improvement.

More recently, **vision transformers** (ViTs) have become the new paradigm for visual recognition
and have made great strides in a broad range of visual recognition tasks (Dosovitskiy et al., 2020;
Wang et al., 2021a; Liu et al., 2021). Several properties of ViT make it a compelling model choice
for visual relational reasoning. First, the **self-attention mechanism** in ViT offers a strong relational
inductive bias, explicitly modeling the relations between input entities. Second, the design of **image
as patches** facilitates the learning of object-centric representations, as evidenced by recent works,
*e.g.* DINO and EsViT (Caron et al., 2021; Li et al., 2021), that demonstrate ViTs trained with
self-supervised learning (SSL) capture objects in the image without label annotations.

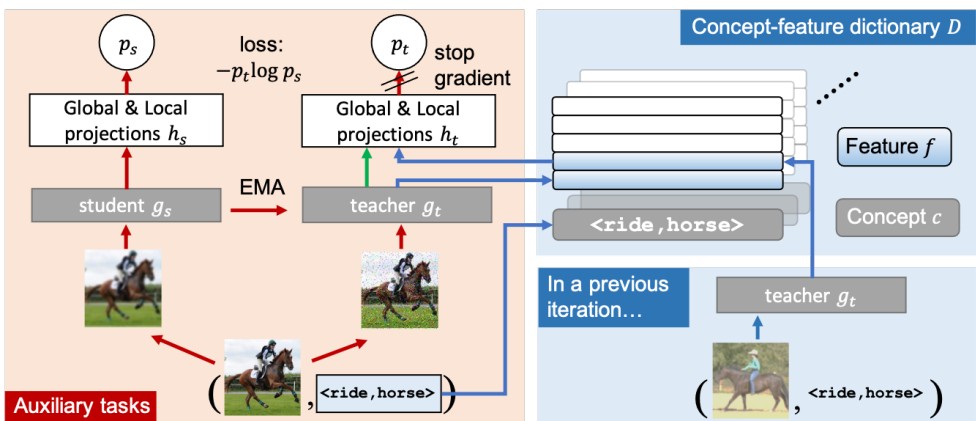

Figure 1: An overview of our method. Red+Green: the learning pipeline of DINO (Caron et al., 2021) and EsViT (Li et al., 2021); Red+Blue: our pipeline. We introduce a *concept-feature dictionary*, where the key is a concept $c$ and its value is a queue of image features $f$ with the same concept, to allow flexible feature retrieval with the concept keys. With the proposed dictionary, we further develop our concept-guided global and local tasks. EMA denotes the exponential moving average.

To investigate the efficacy of the ViT backbone for visual relational reasoning, in particular on systematic generalization, we introduce new systematic splits to canonical benchmarks and compare the ViT backbone with the CNN backbone. Results on GQA show that switching to ViTs in MCAN model (Yu et al., 2019) brings an immediate 11% gain in accuracy. However, the performance gap between the original GQA testing split and the new systematic split remains considerable (15% in accuracy) for both backbones. It suggests that generic ViTs still need to be improved to tackle the reasoning task, especially on systematic generalization. Recent works have shown that neural networks can learn representations with better generalization, by learning certain auxiliary tasks of predicting human-specified concepts (Hill et al., 2020; Koh et al., 2020). A natural question emerges: *can we exploit these concepts to improve the reasoning ability of ViTs?*

**Our approach** is to make better use of concepts (e.g. the labels in the original training dataset) in the ViT training for better relational reasoning. To this end, we first introduce a novel *concept-feature dictionary*, where each key is a concept and its value is a queue of image features with the same concept, as shown in Figure 1. It allows dynamic and flexible training-time image feature retrieval during training. Based on this dictionary, we then augment the canonical ViT training pipeline with two auxiliary tasks: First, to facilitate high-level reasoning about relationships, we design a **global task** that helps cluster images with the same concept together to produce semantically consistent relational representations. Second, to learn better object-centric representations, we develop a **local task** that guides the model to discover object-centric semantic correspondence across images (Liu et al., 2010). Thanks to the plug-and-play feature of our concept-feature dictionary, our auxiliary tasks can be easily incorporated into existing ViT training pipelines without additional input pre-processing. We term the resulting model *concept-guided vision transformer* (or RelViT for short).

We evaluate our method on two standard visual relational reasoning benchmarks: HICO and GQA. Beyond the original independent and identically distributed (I.I.D.) training-testing split, we introduce new systematic splits for each dataset to examine the ability of systematic generalization, *i.e.*, recognizing novel object-relation combinations. Our results show that RelViT significantly outperforms previous approaches. On HICO, it improves the best baseline by 16%, 43%, and 7% on the original non-systematic and two new systematic splits, respectively, as shown in Figure 2. On GQA, it further closes the gap of overall accuracy between models using visual backbone feature only and models using additional bounding box features (obtained from pre-trained object detectors) by 13% and 18% on the two splits. We also show that our method is compatible with various ViT variants and robust to hyperparameters. Finally, our qualitative inspection indicates that RelViT does improve ViTs on learning relational and object-centric representations.

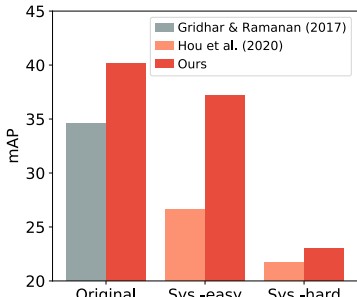

Figure 2: **Results on HICO.** Our method improves the best baseline by 16%, 43%, and 7% on the original non-systematic and two new systematic splits. **Sys.:** systematic.

Our main contributions are summarized as follows:

- We propose RelViT, by incorporating visual relational concepts to the ViT training with the newly-introduced concept-guided global and local auxiliary tasks, where a *concept-feature dictionary* is proposed to enable dynamic and flexible image feature retrieval with the concept keys.

- In extensive experiments on the original non-systematic and new systematic split of the HICO and GQA datasets, we demonstrate the advantages of RelViT over various strong baselines for visual relational reasoning.

- We perform ablation studies on RelViT to show the contributions of its key components, its compatibility to various ViT architectures, and its robustness to hyper-parameters. We provide qualitative results to confirm our improved learning of relational and object-centric representations.

## 2 METHODOLOGY

### 2.1 BACKGROUND

**Vision transformers.** Here we briefly review the architecture of multi-staged ViTs (Dosovitskiy et al., 2020). Given an image $\mathbf{I} \in \mathbb{R}^{H \times W \times C}$, a ViT model $g$ first tokenizes the input into $N$ image tokens (patches) with a resolution of $(T, T)$: $\texttt{tokenize}(\mathbf{I}) = [t_1, \cdots, t_N], t_i \in \mathbb{R}^{T^2 \times C}, N = HW/T^2$, where $(H, W)$ and $C$ denotes the original resolution and number of channel of the image, respectively. Then in each stage, a *patch embedding* and a *multi-head self attention* (MHSA) module is applied to these tokens to produce input for the next stage. The final output of ViT $g(\mathbf{I})$ is a sequence of tokens $[z_1, \cdots, z_N]$ that correspond to the aforementioned input tokens. For global prediction tasks, *e.g.* image categorization, a summary of the input image can be obtained by either inserting an extra `[CLS]` token to the input sequence of image tokens or performing an extra pooling operation over the output tokens (Zhai et al., 2021).

**Self-supervised learning with DINO and EsViT.** Our method is developed upon the recently proposed self-supervised learning (SSL) approach *self-distillation with no labels* (DINO) (Caron et al., 2021) and its follow-up EsViT (Li et al., 2021). As shown in Figure 1, their main idea is to encourage the output consistency between a teacher $g_t$ and a student network $g_s$, parameterized by $\theta_t$ and $\theta_s$, respectively. Given an input image $\mathbf{I}$, both networks map it to a probability distribution $P_t(\mathbf{I}) = h_t(g_t(\mathbf{I}))$ and $P_s(\mathbf{I}) = h_s(g_s(\mathbf{I}))$ via an extra projection head $h(\cdot)$. The teacher and student network will be updated alternatively by following these two rules: (1) For the student network: $\theta_s \leftarrow \arg\min_{\theta_s} \mathcal{L}_{\text{Global}}$, where $\mathcal{L}_{\text{Global}} = -P_t(\mathbf{I}) \log P_s(\mathbf{I})$; (2) For the teacher network, $\theta_t$ is updated using an exponential moving average (EMA) on $\theta_s$: $\theta_t \leftarrow \lambda\theta_t + (1 - \lambda)\theta_s$, where $\lambda$ controls the updating momentum. In practice, multiple views of the input image $\mathbf{I}$ will be generated via data augmentation and the teacher and student networks will receive different views, preventing the task from being trivialized. EsViT further extends the image-level loss $\mathcal{L}_{\text{Global}}$ to patch-level by applying dense SSL (Wang et al., 2021c) for learning correspondence between the different views, enhancing the performance on dense prediction. Readers are encouraged to refer to Caron et al. (2021) and Li et al. (2021) for more details about these two works.

### 2.2 RELVIT

RelViT is a concept-guided ViT that makes better use of the concepts in the ViT training for the improved relational reasoning. In this section, we first introduce a *concept-feature dictionary* to store and retrieve image features with their concept keys. We then augment the canonical ViT training pipeline with two auxiliary tasks: a global level task and a local level task, both are concept-guided by resorting to the concept-feature dictionary. Intuitively, the global task help cluster images with the same concept together to produce semantically consistent relational features, while the local task guides the model to discover object-centric semantic correspondence across images.

**Concept-feature dictionary.** We assume the total number of concepts is $M$, and the set of all concepts $\mathcal{C} = \{c_1, \cdots, c_M\}$. A *concept-feature dictionary* is denoted by $D = \{(c_1, Q_1), \cdots, (c_M, Q_M)\}$, where each concept $c_i$ is associated with a queue $Q_i$ of image features. During training, each image $\mathbf{I}$ may come with multiple concepts, which we denote by $\mathcal{C}_{\mathbf{I}} \subset \mathcal{C}$. For instance, there may exist several human-object interactions in an image from the HICO dataset, each of which may correspond to a concept. As shown in Figure 1, whenever a new image-concept pair $(\mathbf{I}, \mathcal{C}_{\mathbf{I}})$ comes, we uniformly draw a concept code $c$ from $\mathcal{C}_{\mathbf{I}}$, pick up the queue $Q$ from the dictionary that corresponds to $c$, and then retrieve the image feature $f$ from $Q$. Meanwhile, we pass the input image $\mathbf{I}$ to the teacher network $g_t$ to get the new image feature $f' = g_t(\mathbf{I})$, and *enqueue*

it to $Q$. Note that if $Q$ is full already, we first need to *dequeue* the oldest image feature from $Q$. During training, we use the retrieved image feature $f$ for the two auxiliary tasks below, rather than the input image feature $f'$.

Furthermore, the sampling strategy, *i.e.* how to retrieve image feature $f$ from $Q$, plays an important role in the overall performance of our method. We consider the following two sampling strategies:

- *Uniform sampling.* Each image feature is drawn with equal probability from the queue, *i.e.* suppose we have $N$ features in the queue, then the probability of each feature being sampled is $1/N$. This tactic encourages the diversity of the retrieved image features, benefiting the overall performance. However, some older features in the queue may largely fall behind the current model if the teacher network $g_t$ is updated quickly, eliciting unstable training.

- *"Most-recent" sampling.* The sampling probability mass is allocated based on the freshness of image features, and the most recent feature has the highest chance to be retrieved. Specifically, suppose we have $N$ features in the queue $Q$ ($|Q| >= N$). Then for the $i$-th newest feature $f$, we define its weight $w_i = N - i + 1$. Finally, the probability of the $i$-th newest feature being sampled is $w_i / \sum_{j=1}^{N} w_j$. This tactic ensures we retrieve more up-to-date features and thereby stabilizes the learning. But it may hurt the overall performance due to a lack of feature diversity, as the chance of older features being sampled is small.

Note that the feature queue is empty at the beginning of training. In this case, we simply use the input image feature $f'$ for the auxiliary tasks, and also *enqueue* it to $Q$ that corresponds to the concept of the input image. As we can show in the next, now our proposed global and local tasks reduce to DINO (Caron et al., 2021) and EsViT (Li et al., 2021), respectively.

**Concept-guided global task.** Suppose we have two views $\{\mathbf{I}^{(1)}, \mathbf{I}^{(2)}\}$ of an image $\mathbf{I}$, the main idea of our concept-guided global task is to replace $\mathbf{I}^{(1)}$ in the DINO loss (Caron et al., 2021) with the image feature $f$ sampled from the concept-feature dictionary, which becomes

$$\mathcal{L}_{\text{Global}} = -h_t(f) \log h_s(g_s(\mathbf{I}^{(2)})), \tag{1}$$

where $h_t$ and $h_s$ are the projection head of the teacher and student network, respectively, and $g_s$ is the student network. Intuitively, minimizing the global loss is equivalent to encouraging the similarity of any two different image features with the same concept. Hence, it can help produce more semantically consistent relational representations, in particular when the concepts stored in the concept-feature dictionary are themselves relational.

Similar inter-class representation learning techniques have been explored before (Wang et al., 2017; Caron et al., 2018). However, these approaches require a rather complex pre-processing stage, *e.g.* the images have to be split in terms of the concept before training, making them not directly applicable to existing training pipelines. Rather, with our proposed concept-feature dictionary that dynamically saves & retrieves image features from the running storage, our concept-guided global task becomes a plug-n-play task to existing training pipelines.

**Concept-guided local task.** As we mentioned earlier, our concept-guided local task aims at facilitating object-centric learning, by the means of correspondence learning (Liu et al., 2010; Wang et al., 2019). Recent studies have unveiled the possibility of learning correspondence with SSL (Wang et al., 2021c; Li et al., 2021). However, only low-level correspondence between two augmented (*e.g.* rotated) views of an image can be discovered, while the semantic information of objects is missing. To remedy this, we bring concepts to these methods, endowing them the capability of learning semantic correspondence from images.

Specifically, suppose we have two views $\{\mathbf{I}^{(1)}, \mathbf{I}^{(2)}\}$ of an image $\mathbf{I}$, and we also tokenize the image feature into a sequence of $N$ local image tokens. Then at the output of ViT, we obtain $g_t(\mathbf{I}^{(1)}) = [z_1^{(1)}, \cdots, z_N^{(1)}]$ and $g_s(\mathbf{I}^{(2)}) = [z_1^{(2)}, \cdots, z_N^{(2)}]$, where $z$ denotes the local feature. Prior work, such as EsViT (Li et al., 2021), relies on the local features $g_t(\mathbf{I}^{(1)})$ and $g_t(\mathbf{I}^{(2)})$ for the local task. Instead, we replace $g_t(\mathbf{I}^{(1)})$ with the image feature $f$ retrieved from the concept-feature dictionary using the concept of the image $\mathbf{I}$. We then split $f$ into multiple local features, *i.e.* $f = [z_1^{(f)}, \cdots, z_N^{(f)}]$ and our concept-guided local loss becomes

$$\mathcal{L}_{\text{Local}} = -\frac{1}{N} \sum_{i=1}^{N} h_t(z_{j^\star}^{(f)}) \log h_s(z_i^{(2)}), \quad j^\star = \arg\max_{j} \text{CosineDistance}(z_j^{(f)}, z_i^{(2)}), \tag{2}$$

where $h_t(\cdot), h_s(\cdot)$ are the projection heads that map local features to probability distributions[1]. Intuitively, it greedily matches the output between two local regions that have minimal feature distance – bootstrapping the object-level semantic correspondence among images with the same concept.

**Overall loss.** By combining the global and local tasks, we add an auxiliary task loss $\mathcal{L}_{\text{aux}}$ to the main loss $\mathcal{L}_{\text{main}}$ (*e.g.* cross-entropy loss of the reasoning task). The eventual objective is

$$\mathcal{L} = \mathcal{L}_{\text{main}} + \alpha\mathcal{L}_{\text{aux}}, \ \ \mathcal{L}_{\text{aux}} = \mathcal{L}_{\text{Global}} + \mathcal{L}_{\text{Local}}, \tag{3}$$

where a trade-off weight $\alpha$ is added for better flexibility. As we mentioned above, our method will reduce to EsViT, a baseline without concept-guided auxiliary tasks, when we use the current input features $g_t(\mathbf{I}^{(1)})$ instead of $f$ retrieved from our dictionary for computing $\mathcal{L}_{\text{Global}}$ and $\mathcal{L}_{\text{Local}}$.

## 3 EXPERIMENTS

We conduct experiments on two challenging visual relational reasoning datasets: HICO (Chao et al., 2015) and GQA (Hudson & Manning, 2019). Besides their original non-systematic split, we introduce the systematic splits of each dataset to evaluate the systematic generalization of our method. First, we compare our method against various strong baselines (Mallya & Lazebnik, 2016; Girdhar & Ramanan, 2017; Hudson & Manning, 2018a) on visual relational reasoning, as well as state-of-the-art ViTs. Second, we perform the ablation analysis to examine the key components of our method: ViT backbones, concept-feature dictionaries, and auxiliary tasks. Finally, we provide qualitative results to justify the emerging image clustering in terms of concepts and the learned semantic correspondence. Please see more details of all the evaluated tasks in the supplementary material.

### 3.1 MAIN RESULTS I: HUMAN-OBJECT INTERACTION RECOGNITION

**Overview.** HICO (Chao et al., 2015) features the human-object interaction (HOI) recognition, *i.e.* predicting all the possible HOI categories of the input image. It contains 600 HOI categories with 117 unique actions and 80 object classes. The training set includes 38116 images and the test set includes 9658 images. For a fair comparison, we follow the standard practice and mainly focus on those previous methods that do not require extra supervision (Fang et al., 2018) or data (Li et al., 2020b; 2019b; Jin et al., 2021). By default, we choose PVTv2-b2 (Wang et al., 2021b) as the ViT backbone. Regarding the concept-feature dictionary, we use the *"most-recent" sampling* and a queue length $|Q|$ of 10. The trade-off weight $\alpha$ in the overall loss is fixed to 0.1. Other hyper-parameters are inherited from DINO (Caron et al., 2021).

**Systematic split.** The systematic generalization in HICO has been studied before under the name "zero-shot HOI recognition" (Shen et al., 2018). The main idea is to remove some HOI categories from the training set while ensuring all the single actions and objects can still be kept in the remaining HOI categories. We thereby reuse the systematic splits offered by Hou et al. (2020). There are two splits: *systematic-easy*, where only the rare HOI classes are removed from the training set; *systematic-hard*, where only the non-rare HOI classes are removed besides the rare ones. The systematic-hard split contains much fewer training instances and thereby is more challenging.

**Concepts.** In HICO, we simply use the 600 HOI categories as our default concepts. We also report results with other concepts (*e.g.* actions, objects) in the ablation study.

**Results.** In Table 1, we compare our method with several counterparts. The results read that even a simple model with PVTv2-b2 (**25.4M parameters**) backbone can outperform many previous methods using ResNet-101 (**44.7M parameters**) and lots of bell and whistles. This confirms the great potentials of ViTs in visual relation reasoning. By further adding our global and local tasks, we attain 4-6 mAP gain on original and systematic splits. We also observe that EsViT (Li et al., 2021), a recently proposed SSL approach, also outperforms the ViT-only baseline. Therefore, we combine their SSL task and our concept-guided tasks and reach the peak performance (40.12 mAP) on the original HICO split. Although **we do not utilize any extra supervison**, RelViT+EsViT beats the current state-of-the-art Fang et al. (2018) that uses the additional "pose" supervision that does not exist in the HICO dataset. Overall, we raise the results of a fair counterpart (Girdhar & Ramanan, 2017) that only exploits extra bbox supervision (which is included in HICO) by 16% ($34.6 \rightarrow 40.12$) on the original split. For systematic splits, we raise the results of Hou et al. (2020) by 43% (26.65

---

[1]Note that the projection head here is different from DINO's: it works on all output local features. While in DINO, the projection head only works on the summary of input image, *i.e.* the resulting feature after a max-pooling operation or the feature that corresponds to `[CLS]` in the input.

| Method | Ext. superv. | Backbone | Orig. | Systematic-easy | | Systematic-hard | |
|---|---|---|---|---|---|---|---|
| | | | | Full cls. | Unseen cls. | Full cls. | Unseen cls. |
| Mallya & Lazebnik (2016)* | | ResNet-101 | 33.8 | - | - | - | - |
| Girdhar & Ramanan (2017)* | bbox | ResNet-101 | 34.6 | - | - | - | - |
| Fang et al. (2018)* | pose | ResNet-101 | 39.9 | - | - | - | - |
| Hou et al. (2020)† | | ResNet-101 | 28.57 | 26.65 | 11.94 | 21.76 | 10.58 |
| ViT-only | | PVTv2-b2 | 35.48 | 31.06 | 11.14 | 19.03 | 18.85 |
| EsViT (2021) | | PVTv2-b2 | 38.23 | 35.15 | 11.53 | 22.55 | 21.84 |
| RelViT (Ours) | | PVTv2-b2 | 39.4 | 36.99 | 12.26 | 22.75 | 22.66 |
| RelViT + EsViT (Ours) | | PVTv2-b2 | **40.12** | **37.21** | **12.51** | **23.06** | **22.89** |

Table 1: **Results on HICO dataset.** Some methods requires extra supervision. Bbox/Pose means object-detection or pose estimation is needed. All results are reported in mAP. *Results reported in the original papers; †Introduces the systematic split we use in the experiments. **Full cls.**: results reported on all 600 HOI categories; **Unseen cls.**: results reported on the held-out HOI categories from the training set for testing systematic generalization. **Ext. superv.**: extra supervision.

$\rightarrow$ 37.21) on the systematic-easy split and 7% (21.76 $\rightarrow$ 23.06) on the systematic-hard split. Finally, although the gap between systematic and non-systematic split still exists (partly due to the much smaller training set for systematic splits), our method makes significant progress, especially on unseen classes (+12.3 mAP on systematic-hard). This further demonstrates the advantages of our concept-guided ViT in systematic generalization.

## 3.2 MAIN RESULTS II: VISUAL QUESTION ANSWERING

**Overview.** GQA (Hudson & Manning, 2019) is a recent visual question answering (VQA) dataset with a focus on relational reasoning. Each question is also labeled with semantics. By default, it offers both pretrained-CNN grid features and region features obtained through Faster R-CNN (Ren et al., 2015). For counterparts, we focus on fair comparisons and therefore exclude those that require massive vision-language pretraining (Li et al., 2019a). Notably, **we do not use extra supervision, such as scene graph** (Krishna et al., 2016). The RelViT configuration is almost the same as in HICO, except that we apply the *uniform sampling* instead as we find it empirically works better. We employ MCAN-Small (Yu et al., 2019) as our VQA model and the ImageNet1K-pretrained PVTv2-b2 as our vision backbone. The results are reported on the full validation set of GQA.

**Systematic split.** In GQA, we especially examine the facet of *productivity* in systematic generalization, *i.e.* the ability of reasoning with longer reasoning hops (Hupkes et al., 2020). To this end, we exploit the extra semantics label associated with the GQA questions. We observe that the semantics in GQA break down each question into a sequence of "reasoning hops", where a distribution of reasoning hops can be found in Figure 3. See the supplementary material for examples. Therefore, our idea is to exclude questions with longer reasoning hops from the training set. We end up only keeping questions with less than 5 reasoning hops in the training set. We refer to this setting as the systematic split ("Sys.") in the results.

**Concepts.** Inspired by recent research on vision-language pretraining (Tan & Bansal, 2019; Li et al., 2019a; 2020a), we obtain concepts by parsing the questions into keywords. Specifically, we only keep certain verbs, nouns, and adjectives that contain significant meanings (e.g. actions, objects, characteristics, etc), ending up with 1615 concepts. Due to the space limit, readers may find more details on concept parsing in the supplementary material.

**Results.** We report the comparison results on the original and systematic splits in Table 2. The main goal of our experiments on GQA mainly is to verify if our method can help reduce the gap between models using backbone features only and models using additional bbox features (with dense object detectors). Besides, we also examine to which extent our method can improve systematic generalization. Firstly, we observe that using ViT can largely alleviate the aforementioned gap (51.1 $\rightarrow$ 56.62), suggesting that the object-centric representations emerge in ViTs. It implies the potential of using ViTs in eliminating the need for external object detectors. By further adding our proposed auxiliary tasks, we achieve the peak performance and raise the results of MCAN-Small w/o bbox features by 13% (51.1 $\rightarrow$ 57.87) on the original split and 18% (30.12 $\rightarrow$ 35.48) on the systematic split. **Without any detection pretraining or bbox features**, our method achieves very close results to MCAN-Small w/ bbox features on both two splits. The additional results in appendix demonstrate

| Method | Bbox feat.* | Backbone | Orig. | Sys. |
|---|---|---|---|---|
| BottomUp (2018) | ✓ | ResNet-101 | 53.21 | - |
| MAC (2018b) | ✓ | ResNet-101 | 54.06 | - |
| MCAN-Small (2019) | ✓ | ResNet-101 | 58.35 | 36.21 |
| MCAN-Small (2019) | | ResNet-101 | 51.1 | 30.12 |
| ViT-only | | PVTv2-b2 | 56.62 | 31.39 |
| EsViT (2021) | | PVTv2-b2 | 56.95 | 31.76 |
| RelViT (Ours) | | PVTv2-b2 | 57.87 | 35.48 |

Table 2: **Results on GQA dataset.** All results are reported in overall accuracy. *With extra Faster R-CNN bbox features.

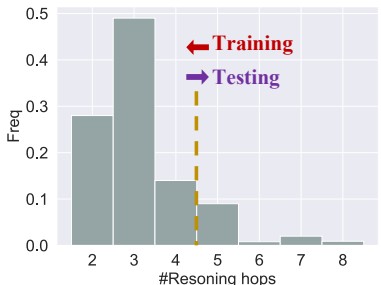

Figure 3: Histogram of reasoning hops over GQA training questions.

that the marginal gap could be further eliminated if we apply larger backbone models (PVTv2-b2 has much fewer parameters than ResNet-101).

### 3.3 WHY DO OUR AUXILIARY TASKS WORK?

The results in the previous section suggest that RelViT outperforms its counterparts on the challenging relational reasoning tasks. Now we would like to provide more insights into our method design by answering the question: why do our auxiliary tasks work? To this end, we perform a diverse set of analyses on accessing the impact of key components in RelViT . We also qualitatively justify the intuitions of two auxiliary tasks. These results are reported on the HICO dataset.

#### 3.3.1 ABLATION STUDY

**Different ViT architectures.** The first aspect we examine is the ViT architecture. Besides the default choice on PVTv2-b2, we test our method with the original ViT-S/16 (Dosovitskiy et al., 2020) and another prevalent architecture Swin-Small (Liu et al., 2021). The results are presented in Figure 4a and Figure 4b, respectively. These two architectures can both benefit from our auxiliary tasks and we have similar advantages over counterparts as in the default setting, which confirms our compatibility to various ViT variants. Full quantitative results are provided in the supplementary.

**Implementation of concept-feature dictionary.** We conduct ablations on three facets of concept-feature dictionary: the choice of concepts, sampling tactics, and the size of queue $|Q|$. In Figure 4c, we compare three different concept choices: actions, objects, and HOIs with our best model. The results suggest that all three choices can bring improvement to the baseline without any feature queue (denoted as "None") while using HOIs and objects brings larger improvement. We hypothesize that the proposed auxiliary tasks need more "delicate" concepts to guide the ViT training but actions in HICO tend to be vague and even ambiguous (Shen et al., 2018). Therefore, albeit the consistent advantages of our method in terms of different concept selections, a careful design of concept space could still be pivotal to achieve the peak performance in relational reasoning.

Furthermore, we show the interplay between sampling strategies and queue size $|Q|$ in Figure 4d. Interestingly, $|Q|$ has a much smaller impact on the performance with the *"most-recent" sampling* than that with the *uniform sampling*. As we mentioned in Section 2.2, the *uniform sampling* could help with more diverse features but could also elicit unstable training. Larger $|Q|$ makes the two consequences in the *uniform sampling* more prominent, thus causing worse performance when stable training is the bottleneck (*e.g.* in a small dataset like HICO). Rather, the *"most-recent" sampling* can be less sensitive to $|Q|$ as only the recent features could be sampled even when $|Q|$ is large.

**Auxiliary tasks.** In Figure 4e, we show the results of only adding our global or local task in $\mathcal{L}_{aux}$. Surprisingly, just using the local task is enough to deliver competitive results in the HICO task. This suggests that the real bottleneck in ViTs seems to be better object-centric representations, as our local task is designed for this. Nonetheless, adding our global task can still elicit clear advantages over other counterparts that do not exploit concept-guided learning.

**Robustness to $\alpha$.** We sweep the trade-off weight $\alpha$ from 0.02 to 0.5 and report the results in Figure 4f, where **solid** and **dash** represent our method and the baseline, respectively. It is observed that adding the proposed auxiliary tasks always achieves better performances than the baseline, indicating our method is robust to hyper-parameters. Moreover, the improvements become slightly more significant when $\alpha$ is relatively large (but not too large). The peak performances in different splits all appear around $\alpha = 0.1$, which we thus use as our default choice.

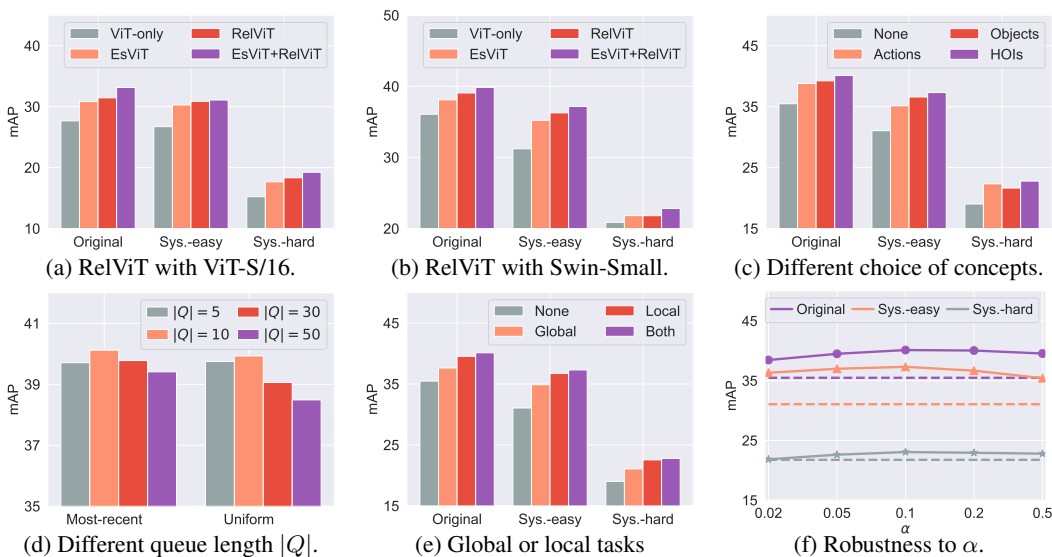

Figure 4: **Ablation study on HICO.** We investigate the impact of ViT architectures, implementation of concept-feature dictionary, auxiliary tasks, and the weight $\alpha$ on the performance of our method. **Sys.**: systematic.

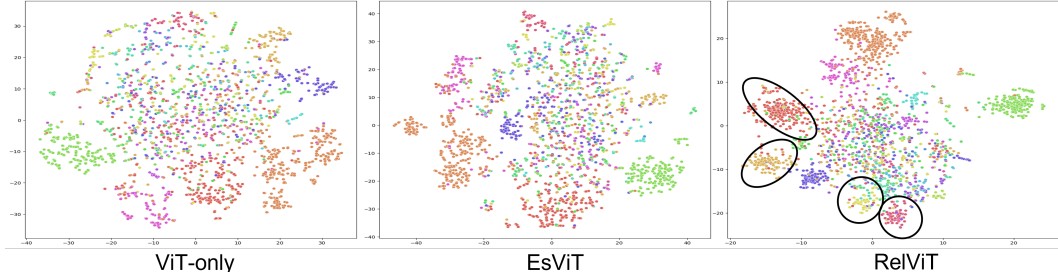

Figure 5: Visual illustrations of image features against HOI categories on the HICO test set via t-SNE. We compare the features obtained by ViT without any auxiliary task (ViT-only), ViT with non-concept auxiliary tasks (EsViT), and RelViT. Besides those clusters that are identified with the other two baselines, **clusters that can only be identified with RelViT are highlighted.**

### 3.3.2 QUALITATIVE INSPECTION

**Features vs. concepts.** To further justify whether our global task can truly facilitate the learned representation to be more relational, we illustrate the learned output features (max-pooling on all the output tokens) by t-SNE visualization in Figure 5. Different colors correspond to different HOI categories, *i.e.* the concepts we used in RelViT. The results read that more clusters can be identified over the image features extracted by RelViT; therefore our concept-guided global task can encourage the learned features to be more discriminative regarding the relational concepts than the baselines.

**Semantic correspondence.** We also probe the learned semantic correspondence that could be encouraged by our local task, by intuition. We aim at comparing the correspondence extracted from a model trained with different auxiliary tasks, *i.e.* no auxiliary task, no-concept auxiliary tasks, and our auxiliary tasks. We consider two settings: 1) semantic setting (two images that belong to the same concept, *e.g.* both contains a cat), and 2) non-semantic setting (two views of the same image). Results in Figure 6 highlight the high-similarity matches. Although our method and non-concept baseline (EsViT) both work fine in the non-semantic setting, our method can identify the semantic correspondence on more objects thanks to the concept guidance. Not surprisingly, baseline w/o any auxiliary task (ViT-only) performs the worst as it may suffer from over-smoothness (Gong et al., 2021) and lose all the meaningful spatial information after fine-tuning on the target task.

## 4 RELATED WORK

**Systematic generalization in visual reasoning.** Systematic generalization (Hupkes et al., 2020; Bahdanau et al., 2018) characterizes to which extent a learning agent can identify and exploit the

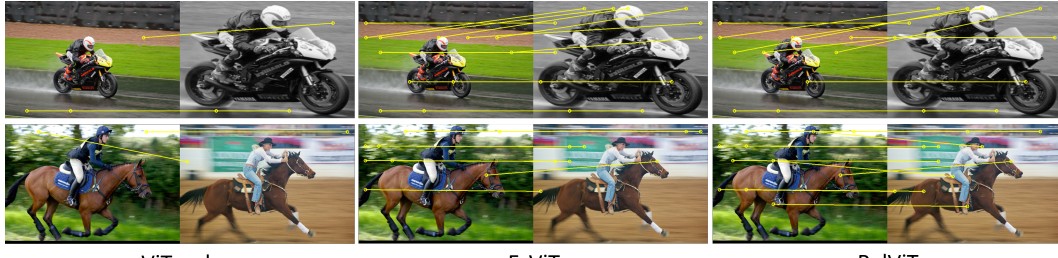

ViT-only          EsViT          RelViT

Figure 6: Visualization of correspondence. The correspondence is extracted between two views of the same image (upper) and two images that belong to the same concept (lower), using the learned model on HICO. **RelViT can extract correspondence on more objects in the two images (semantic correspondence) setting**. Best viewed on screen.

underlying entities and relations of the training data, and generalize to novel combinations and longer reasoning hops. There has been extensive research on inspecting and tackling systematic generalization in visual reasoning (Johnson et al., 2017; Kim & Mnih, 2018; Higgins et al., 2016; Kuhnle & Copestake, 2017). However, most of them only focus on controlled and synthetic domains (Ruis et al., 2020; Zhang et al., 2019; Barrett et al., 2018; Xie et al., 2021; Nie et al., 2020; Jiang et al., 2022), while the open-ended real-world domains are largely neglected with very few exceptions (Shen et al., 2018; Teney et al., 2020). In this paper, we tackle systematic generalization in visual relational reasoning with natural images, thereby filling the gap between synthetic and real domains.

**Object-centric and relational representations.** Many seminal research reveals that ML models can benefit from object-centric and relational representations with better sample efficiency and generalization (Farhadi & Sadeghi, 2013; Ding et al., 2020; Mrowca et al., 2018). However, obtaining such representations from unstructured inputs, *i.e.* raw images, still remains challenging (Greff et al., 2019; Locatello et al., 2020; 2019; Yu et al., 2021). Prevalent approaches adopt a latent variable model to explicitly infer the foreground-background split as well as objects & relations (Eslami et al., 2016; Lin et al., 2020; Zhu & Mumford, 2007), while recent findings suggest that they can be an emerging property of transformers trained with self-supervised objectives (Caron et al., 2021; Li et al., 2021). Our goal aligns better with the later regime, as it enables implicit representations and thus could be more versatile and efficient. A key difference is that these methods do not exploit concepts in reasoning benchmarks, making them less capable of learning semantic representations.

**Self-supervised learning for ViTs.** The recent resurgence on self-supervised learning (SSL) of image models has delivered impressive results on many few-shot or zero-shot tasks (Oord et al., 2018). From a high-level view, these approaches can be categorized into *contrastive* (He et al., 2020; Chen et al., 2020) and *non-contrastive* (Chen & He, 2021). However, not all SSL avenues work well with vision transformers (ViTs) and some delicate design may be needed. Caron et al. (2021) found their non-contrastive learning objective (DINO) manifested better quantitative results and emerging properties on ViTs. Chen et al. (2021) brought similar results on contrastive SSL. Li et al. (2021) further introduced patch-level SSL objective to ViTs for dense prediction tasks. In this paper, instead of proposing a new SSL approach, we make better use of concepts for ViT training, which can be directly applied to the existing SSL objectives for the improved visual reasoning.

## 5    CONCLUSION

In this paper, our goal is to seek a better inductive bias for visual relational reasoning, especially on real-world data. We found ViTs to be a promising candidate due to their potential on relational reasoning, object-centric learning, and systematic generalization. We further presented RelViT, a simple yet efficient method for exploiting concepts in the visual relational reasoning tasks to boost the performances of ViTs. In specific, we proposed two auxiliary tasks in RelViT : a global task for semantically consistent relational representation, and a local task for learning object-centric semantic correspondence. These two tasks are made possible through the use of our proposed concept-feature dictionary. RelViT largely outperforms other counterparts on two challenging visual relational reasoning benchmarks. While we mainly focus on extending ViTs to visual reasoning using auxiliary tasks, further exploration of combining our work with architectural modification over ViTs to enable better generalization could be a new direction for future work.

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

## A A FORMAL DESCRIPTION OF LEARNING IN RELVIT

Algorithm 1 formally depicts the execution flow of RelViT.

---

**Algorithm 1** RelViT: Concept-guided Vision Transformer

---

**Input:** A set of training images with concepts $\{(\mathbf{I}_1, C_1), \cdots\}$, an image augmentation function $\mathtt{aug}(\cdot)$, momentum update factor $\lambda$, loss weight $\alpha$, a concept-feature dictionary $D$, teacher and student ViT $g_t$ and $g_s$, parameterized by $\theta_t$ and $\theta_s$, respectively.

1: **for** $(\mathbf{I}_i, C_i)$ in $\{(\mathbf{I}_1, C_1), \cdots\}$ **do**
2:      $\mathbf{I}_i^{(1)}$, $\mathbf{I}_i^{(2)} = \mathtt{aug}(\mathbf{I}_i)$, $\mathtt{aug}(\mathbf{I}_i)$
3:      Uniformly draw a concept code $c \sim C_i$.
4:      Retrieve $Q$ from $D$ with $c$.
5:      **if** $Q$ is not empty **then**
6:          Sample feature $f \sim Q$, following some sampling tactics.
7:          $\mathcal{L}_{\text{aux}} = \mathcal{L}_{\text{Global}}(f, \; g_s(\mathbf{I}_i^{(2)})) + \mathcal{L}_{\text{Local}}(f, \; g_s(\mathbf{I}_i^{(2)}))$
8:          Insert feature $g_t(\mathbf{I}_i^{(1)})$ into $Q$; if it is full, remove the oldest feature.
9:      **else**
10:         $\mathcal{L}_{\text{aux}} = \mathcal{L}_{\text{Global}}(g_t(\mathbf{I}_i^{(1)}), \; g_s(\mathbf{I}_i^{(2)})) + \mathcal{L}_{\text{Local}}(g_t(\mathbf{I}_i^{(1)}), \; g_s(\mathbf{I}_i^{(2)}))$
11:      **end if**
12:      Update $\theta_s$ with the loss function $\mathcal{L} = \mathcal{L}_{\text{main}} + \alpha\mathcal{L}_{\text{aux}}$.
13:      Update $\theta_t$ using an EMA: $\theta_t \leftarrow \lambda\theta_t + (1 - \lambda)\theta_s$.
14: **end for**

---

## B EXTRA DETAILS ON RELVIT

### B.1 INPUT PIPELINE

We adopt the following data augmentation pipeline for the generating the additional views for our two auxiliary tasks

1. Randomly crop and resize the image into $(224, 224)$ with scale ratio $(0.2, 1.0)$;

2. Randomly jitter the color of the image on brightness, contrast saturation and hue with probability of $(0.4, 0.4, 0.4, 0.1)$, respectively;

3. Randomly turn the image into gray scale with probability $0.2$;

4. Randomly apply Gaussian blur with kernel size 23 and sigma $(0.1, 2.0)$ and probability $0.5$;

5. Randomly flip the image horizontally.

Notably, we apply a random crop operation to ensure that all the input images for our auxiliary tasks contain the same number of patches.

### B.2 HYPER-PARAMETERS AND BASELINES

Table 3 summarizes the hyper-parameters used by RelViT. We inherit most of the parameters from DINO (Caron et al., 2021).

Table 3: Hyperparameters for RelViT.

| Parameter | Value |
|---|---|
| Optimizer | AdamW with epsilon $1e^{-1}$ (HICO) / $1e$-5 (GQA) |
| Gradient clipping norm | No grad clipping (HICO) / 0.5 (GQA) |
| Base learning rate | $1.5e^{-4}$ (HICO) / $3e^{-5}$ (GQA) |
| Learning rate schedule | 0.1 scale with milestones $[15, 25]$ (HICO) / $[8, 10]$ (GQA) |
| Batch size | 16 (HICO) / 64 (GQA) |
| Total training epochs | 30 (HICO) / 12 (GQA) |
| Temperature $\tau$ in DINO loss | 0.04 for teacher and 0.1 for student, we don't use schedule. |
| Momentum $m$ for teacher | 0.999 |
| Center $m$ for center features | 0.9 |
| Sampling method | *"most-recent"* (HICO) / *uniform* (GQA) |
| Queue size $|Q|$ | 10 |

Table 4 summarizes the key details about the loss implementation of different baselines and RelViT
.

Table 4: Key details about the loss implementation in baselines and RelViT .

| | $\mathcal{L}_{\text{Global}}$ | $\mathcal{L}_{\text{Local}}$ | Compare `student(aug(img))` with |
|---|---|---|---|
| DINO | x | | `teacher(aug(img))` |
| EsViT | x | x | `teacher(aug(img))` |
| RelViT | x | x | `queues[concept(img)].pop()` |
| RelViT + EsViT | x | x | `teacher(aug(img))` and `queues[concept(img)].pop()` |

## C  EXTRA DETAILS ON THE DATASETS

### C.1  HICO

#### C.1.1  ORIGINAL AND SYSTEMATIC SPLITS

Besides the official training/testing split, we adopt the splits for systematic generalization presented
in (Hou et al., 2020). It offers two splits that follow different strategies to select held-out HOI
categories. **Systematic-easy** only select *rare* HOI categories (with less than 10 training samples),
while **Systematic-hard** select *non-rare* categories instead. Therefore, the training set of **Systematic-hard** will contain much fewer samples and become more challenging. Some basic statistics of these
training/testing splits can be found in Table 5.

| Splits | #Training samples | #Training HOIs | #Testing samples | #Testing HOIs |
|---|---|---|---|---|
| Original | 38118 | 600 | 9658 | 600 |
| Systematic-easy | 37820 | 480 | 9658 | 600 |
| Systematic-hard | 9903 | 480 | 9658 | 600 |

Table 5: Statistics of the splits of HICO dataset.

#### C.1.2  IMPLEMENTATION OF $\mathcal{L}_{\text{main}}$

In HICO, there might be multiple HOIs for a single image. We, therefore, formulate the HOI pre-diction task as a multi-class classification problem. Specifically, the model makes 600 binary clas-sifications and $\mathcal{L}_{\text{main}}$ in equation 3 is a binary cross-entropy loss.

## C.2  GQA

### C.2.1  ORIGINAL AND SYSTEMATIC SPLITS

We introduce a systematic split for the GQA dataset that is based on reasoning hops. Specifically, we remove those questions that have more than 4 reasoning hops from the training set. Some basic statistics of these training/testing splits can be found in Table 6.

| Splits | #Training samples | #Testing samples |
|---|---|---|
| Original | 943000 | 132062 |
| Systematic | 711945 | 32509 |

Table 6: Statistics of the splits of GQA dataset.

### C.2.2  REASONING HOPS

Since all the questions and answers in the GQA dataset are synthetic, it additionally provides "semantics" that characterizes the reasoning procedure that generates the answer from a question and a visual scene. These semantics are composed of multiple "reasoning primitives" that act like functions: receiving arguments and generating output for the next reasoning step. It is believed they can reflect whether a question will require complex multi-hop reasoning – a pivotal angle of systematic generalization. Therefore we develop our systematic split with it. Table 7 provides a few examples on semantics.

| Question | Semantics (Reasoning hops) |
|---|---|
| Is the pizza with the pepper small and covered? | `relate([0], pizza, with, s(1130674));`
`filter([1], pizza);`
`verify([2], covered);`
`verify_size([2], small);`
`and([3,4]);` |
| Do you see any tablecloths or dressers? | `select([], dreser);`
`exist([0], ?);`
`select([], tablecloth);`
`exist([2], ?);`
`or([1, 3], ?);` |
| Are there microwave ovens to the right of the appliance near the window? | `select([], window);`
`relate([0], appliance, near, s(1297947))`
`relate([1], microwave, right, s(1297947));`
`exist([2], ?);` |

Table 7: Examples of semantics (reasoning hops) in GQA dataset.

### C.2.3  CONCEPT PARSING

We obtain the concepts in the GQA dataset by parsing the questions into word tokens. Specifically, we construct a set of concepts that contain nouns, verbs, and adjectives that are with significant meaning. We also manually filter some ambiguous words from this set. The resulting concept set contains 1615 concepts.

We use the python nltk package to process the question. The parsing procedure starts with part-of-speech tagging, where we only keep nouns (NN), verbs (VB) and adjectives (JJ). Then we lemmatize the remaining words to obtain the minimal form of them. Finally, we remove those that do not present in the pre-selected concept list. Additionally, we skip questions with "No" as the answer as the question may be unrelated to the image. We provide the statistics of the concepts in GQA in Table 8. The number of associated questions of all the 1615 concepts and a histogram on the number of concepts for each question is presented in Figure 7a and Figure 7b, respectively.

| Item | Value |
|---|---|
| Questions without concept | 166217 out of 943000 (17.6%) |
| Concepts without any question | 14 |
| Concepts with $< 10$ questions | 209 |
| Averaged #questions per concept | 1030.9 |
| Median #questions per concept | 106 |
| Top 20 concepts and their #associated questions | man 52295 |
| | animal 44070 |
| | furniture 36523 |
| | white 33141 |
| | front 30779 |
| | person 28751 |
| | vehicle 26133 |
| | woman 25769 |
| | bottom 22624 |
| | black 22517 |
| | device 21962 |
| | food 19683 |
| | fence 19172 |
| | chair 18872 |
| | table 18649 |
| | hold 18090 |
| | shirt 16483 |
| | blue 15434 |
| | car 14838 |

Table 8: Statistics of concepts in GQA training set.

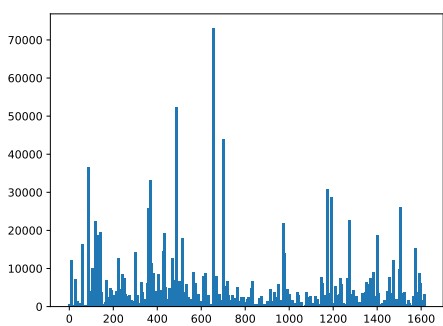

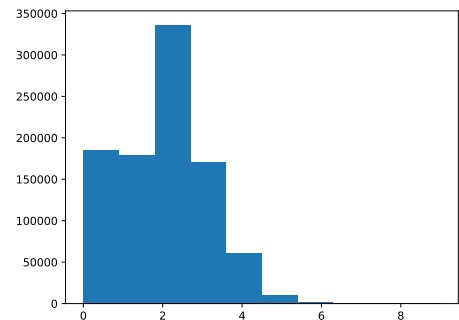

(a) Histogram of number of questions per concept.  (b) Histogram of number of concepts per question.

Figure 7: Histograms of concepts in GQA training set.

### C.2.4 IMPLEMENTATION OF $\mathcal{L}_{\mathrm{main}}$

GQA is formulated as a classification problem, *i.e.* the learner needs to select an answer from the pre-defined answer set; thus $\mathcal{L}_{\mathrm{main}}$ in equation 3 is a cross-entropy loss.

## D FULL QUANTITATIVE RESULTS ON ABLATION STUDIES

We provide the full quantitative results of the ablation studies in Table 9.

| Method | Orig. | Sys.-easy | Sys.-hard |
|---|---|---|---|
| ViT-only | 27.67 | 26.72 | 15.23 |
| EsViT | 30.83 | 30.28 | 17.67 |
| RelViT | 31.45 | 30.88 | 18.33 |
| EsViT+RelViT | 33.15 | 31.09 | 19.24 |

(a) RelViT with ViT-S/16

| Method | Orig. | Sys.-easy | Sys.-hard |
|---|---|---|---|
| ViT-only | 36.08 | 31.22 | 20.88 |
| EsViT | 38.11 | 35.22 | 21.82 |
| RelViT | 39.07 | 36.27 | 21.81 |
| EsViT+RelViT | 39.86 | 37.17 | 22.82 |

(b) RelViT with Siwn-Small

| Concepts | Orig. | Sys.-easy | Sys.-hard |
|---|---|---|---|
| None | 35.48 | 31.06 | 19.03 |
| Actions | 38.8 | 35.14 | 22.34 |
| Objects | 39.24 | 36.59 | 21.65 |
| HOIs | 40.12 | 37.31 | 22.79 |

(c) Different choice of concepts

| $|Q|$ | Most-recent | Uniform |
|---|---|---|
| 5 | 39.71 | 39.75 |
| 10 | 40.12 | 39.93 |
| 30 | 39.78 | 39.06 |
| 50 | 39.41 | 38.49 |

(d) Different queue length $|Q|$

| Tasks | Orig. | Sys.-easy | Sys.-hard |
|---|---|---|---|
| None | 35.48 | 31.06 | 19.03 |
| Global | 37.63 | 34.88 | 21.07 |
| Local | 39.54 | 36.74 | 22.55 |
| Both | 40.12 | 37.31 | 22.79 |

(e) Global or local tasks

| $\alpha$ | Orig. | Sys.-easy | Sys.-hard |
|---|---|---|---|
| 0.02 | 38.45 | 36.32 | 21.85 |
| 0.05 | 39.49 | 36.99 | 22.62 |
| 0.1 | 40.12 | 37.31 | 23.06 |
| 0.2 | 40.04 | 36.67 | 22.94 |
| 0.5 | 39.54 | 35.42 | 22.79 |

(f) Robustness to $\alpha$

Table 9: Full quantitative results (on full class of HICO) of the ablation studies.

# E ADDITIONAL RESULTS

## E.1 RELVIT WITH LARGER BACKBONE MODELS

As we mentioned in Section 3.1, the ViT backbone we use (PVTv2-b2) only has **25.4M** parameters, even less than the commonly-used ResNet-101 (**44.7M** parameters). Therefore, we testify RelViT with larger state-of-the-art ViT backbones: PVTv2-b3 (**45.2M** parameters) and Swin-base (**88M** parameters) (Liu et al., 2021) and provide the results on HICO and GQA below:

Table 10: Results with larger ViT models on HICO.

| HICO mAP | Fang et al. (2018) | RelViT + EsViT (PVTv2-b2) | RelViT + EsViT (PVTv2-b3) | RelViT + EsViT (Swin-base) |
|---|---|---|---|---|
| Original | 39.9 | 40.12 | 42.61 | **43.98** |
| Systematic-easy | - | 37.21 | 39.92 | **42.04** |
| Systematic-hard | - | 23.06 | 25.56 | **28.36** |

Table 11: Results with larger ViT models on GQA.

| GQA overall accuracy | MCAN-Small (w/ bbox) | RelViT (PVTv2-b2) | RelViT (PVTv2-b3) | RelViT (Swin-base) |
|---|---|---|---|---|
| original | 58.35 | 57.87 | 61.41 | **65.54** |
| systematic | 36.21 | 35.48 | 36.25 | **37.51** |

## F   EXTRA DISCUSSION ON THE RELATED WORK

### F.1   DISCUSSION ON THE MEMORY BANK MECHANISM

Intuitively, the idea of using the concept-feature dictionary to the ViT training could be similar to the memory bank mechanism in MoCo (He et al., 2020), where the features are stored in a queue for replaying later. However, the difference is also clear: we have multiple queues that are indexed by concept codes while MoCo only has a single queue. Similar use of memory bank can also be found in Wu et al. (2018); Tian et al. (2020) but they follow MoCo, and therefore it is used for providing negative samples when computing the self-supervised contrastive learning loss. Rather, our concept-feature dictionary is designed to make better use of the concept supervision via our concept-guided global and local losses to improve the performance on visual relational reasoning.

### F.2   COMPARISON TO MAC (HUDSON & MANNING, 2018B)

Here we highlight the difference between our concept-feature dictionary and the knowledge base in MAC (Hudson & Manning, 2018b): The knowledge base in MAC is used during a single VQA reasoning pass (i.e. it will be cleared & initialized with the new image features (visual knowledge) whenever a new `<image, question>` pair comes), and thus is used **in both the training and testing time** for the VQA reasoning. However, the concept-feature dictionary in RelViT is used to store & retrieve features according to the concept of the current input image and help compute our local and global losses that encourage learning better representations. Therefore, we use it **in the training time only** as these two losses won't be computed & optimized during testing.

