# OpenReview forum: "RelViT: Concept-guided Vision Transformer for Visual Relational Reasoning"
_ICLR.cc/2022/Conference — ICLR 2022 Poster_

### Official Review · Reviewer_RiEe · 2021-10-25

**Correctness:** 3
**Technical Novelty And Significance:** 2
**Empirical Novelty And Significance:** 3
**Recommendation:** 6
**Confidence:** 4

**Details Of Ethics Concerns:**

No.

**Main Review:**

Pros:
+ Non-trivial motivation to marry the ViT and contrastive learning to advance visual relation reasoning.
+ Impressive improvements upon the HICO and GQA in the common setting and generalization setting.
+ Detailed ablations of the proposed method.
+ Discussion and results about the ViT and visual relation reasoning are inspiring.

Cons:
- Novelty: Model and auxiliary follow the existing works like DINO, EsViT, Liu et.al. 2010, Wang et al., 2021c, etc. Though the feature dictionary is somewhat new, it is still essential to fully discuss the difference between it and the works including memory modules like Wu et al., 2018, Tian et al., 2020, even they are not based on Transformer.
- Following the above point, this work proposes a useful pipeline by following the existing works, the empirical contribution is more visible than the insight novelty.
- Compared with EsViT, the improvements are much more marginal, due to the less design difference.
- Using the images from HICO, the HOI concepts are complex and suitable for relation reasoning. But HOI images usually contain multiple HOI human-verb-object pairs in one image. For example, in HICO, one image usually has more than 2 h-o pairs. So, during the training, what is the effect of the overlapped concepts in contrastive learning, e.g., the largely overlapped pos-neg concepts. Thus, if applying the proposed work to the instance-level concepts (one human-verb-object pair bbox), what would happen?
- The comparison about the "complex pre-processing stage". This point seems unconvincing. In my opinion, these previous settings are a kind of design not for "efficiency". And if the proposed method has an obvious advantage on efficiency, supportive results should also be provided. So does the claim about the long-tailed data distribution. In HICO, the long-tailed distribution is very severe. Thus, a discussion about the alleviation of the long-tailed bias would make this work more solid.
- "reasoning hops can be found in Table 3." --> "Figure 3"?
- "using ViTs in eliminating the need for external object detectors", this discussion is very interesting and inspiring, could you please give more analyses combining the discoveries of the proposed model?

**Summary Of The Paper:**

This work studies visual relationship reasoning and proposes a model based on the recent contrastive learning methods DINO and EsViT. Compared with the two predecessors, a feature dictionary is proposed to provide the online updated concept features for the teacher model instead. Correspondingly, two auxiliary tasks are introduced to drive semantic concept learning. Experiments on two large benchmarks show the efficacy of the proposed method. And extensive ablations are also made to probe the model components.

**Summary Of The Review:**

Overall, this is an interesting work to dig into the performance of ViT on visual relation reasoning and semantic concept learning. Some non-trivial contributions are made to reveal the effect of contrastive learning designs. And the results on the two benchmarks look promising. However, though studying a very different problem, the method contribution seems weak compared to the previous works. Thus, I think the rating of 6 is suitable for this version. I will adjust my rating according to the response and the discussions from the authors and the other reviewers.

---

### Official Review · Reviewer_dCw4 · 2021-11-01

**Correctness:** 3
**Technical Novelty And Significance:** 3
**Empirical Novelty And Significance:** 3
**Recommendation:** 8
**Confidence:** 4

**Main Review:**

Strengths:
- Well-organized paper, it's clear which previous works the method is built upon and what the novel contributions are
- A relatively simple implementation based on queues of previously-processed images is shown to improve empirical results on two challenging datasets (state-of-the-art mAP in HICO recognition, good results in GQA).
- Well-detailed appendix, all the info about data pipeline and training hyperparameters should ensure full reproducibility of the results
- Well-designed ablation studies that cover most of the questions that one can have while reading the description of the method

Weaknesses:
- The paper lacks a proper comparison with MoCo which is only mentioned briefly in section 4 (K. He, H. Fan, Y. Wu, S. Xie, and R. Girshick, “Momentum Contrast for Unsupervised Visual Representation Learning,”, https://openaccess.thecvf.com/content_CVPR_2020/html/He_Momentum_Contrast_for_Unsupervised_Visual_Representation_Learning_CVPR_2020_paper.html
). In this paper, in fact, many of the concepts of RelViT are already present, almost with the same names, e.g. "At the core of our approach is maintaining the dictionary as a queue of data samples" (MoCo, section 3.2). The main differences are: MoCo uses a single queue and not many concept-based queues, the queue contains images and not features, teacher features are computed on the fly using the current parameters for the teacher.
- The additional losses in the method are described as weakly-supervised, e.g. "we use concepts as a source of weak supervision" in section 4. I think this might be a misrepresentation of the method. In fact, for both use cases demonstrated in the paper, the concepts were extracted from the fully-supervised training labels of each dataset. For example, the basic training signal in HICO already includes the concepts used to build the dictionary, therefore it can not be represented as a form of weak supervision. I suggest rephrasing the relevant paragraphs so that it's clear that RelViT is not a weakly-supervised method but it makes better use of the strong labels present in a dataset.
- Since the model is applied to relationship detection, I suggest adding experiments on VRD (C. Lu, R. Krishna, M. Bernstein, and L. Fei-Fei, “Visual Relationship Detection with Language Priors” http://arxiv.org/abs/1608.00187) and Unrel (J. Peyre, I. Laptev, C. Schmid, and J. Sivic, “Weakly-supervised learning of visual relations” http://arxiv.org/abs/1707.09472), which are two popular datasets for visual relationship detection
- One ablation study on the concept-based queues is missing: using a single queue of old images. This would verify whether a replay buffer alone (akin to keeping track of multiple teachers) already brings an improvement over using the features of the current teacher only. However, maybe this is already present and corresponds to the "None" label in the plot, it's not very clear from the text whether None means no queue at all or a single non-concept-based queue.

Questions:
- $L_{main}$ is left unspecified in section 2.2, it would help if the loss was detailed in full for the tasks used in the experiments. At a first read, I missed the fact that HICO is used for a recognition task (just predict the list of relationships present in an image) and not for detection (detect object bounding boxes and predict relationships). It might be my bias because I'm used to seeing HICO for detection, but also because $L_{main}$ for HICO is not properly specified.
- For the local task, patch-based features from the student's image and the teacher's queued image are greedily matched in the loss. Is the match guaranteed to be 1:1? Is this desirable or not desirable? Can you add a discussion about this?
- Is "RelViT" in figure 6 only RelViT or RelViT+EsVit? If it's only RelViT, without the local loss, why is it so good at matching local regions?

Minor points:
- Equation numbers would help reference specific parts of the paper
- From section 2.2 it's not clear which part of the method corresponds to the experiment rows in table 1: EsViT, RelViT, RelViT+EsViT. The confusion arises also in other parts of the paper. I suggest making the similarities and differences explicit, e.g. with the table below

|              | $L_{global}$ | $L_{local}$ | Compare `student(aug(img))` with |
|--------------|--------------|-------------|----------------------------------|
| DINO         | x            |             | `teacher(aug(img))`              |
| EsViT        | x            | x           | `teacher(aug(img))`              |
| RelViT       | x            |             | `queues[concept(img)].pop()`     |
| RelVit+EsViT | x            | x           | `queues[concept(img)].pop()`     |

**Summary Of The Paper:**

The paper extends the training objective of EsViT to make it better suited to visual relationship detection. This is done by replacing the teacher's features used in both local and global losses with features taken from a queue. The queue is populated with teacher features extracted from previous images. Furthermore, the queue is partitioned by "concepts", e.g. HOI labels or VQA concepts, so that the current student image can be compared with a previous teacher image that contains the same concept instead of any random image.

The novel queue-based formulation is tested in various experiments, both for visual relationship detection and visual question answering. Individual components of the method are ablated and discussed, namely the dictionary of queues, the EsViT-style local loss, and other hyperparameters.

**Summary Of The Review:**

This is a good paper in terms of ideas, writeup and experiments. My suggestions and critics would require just minor adjustments to the paper that can be done in the rebuttal phase. Unless major issues arise in the discussion with other reviewers, e.g. the method is not as novel as I thought due to related works that I am not aware of, my recommendation is to accept.

---

### Official Review · Reviewer_fbWL · 2021-11-02

**Correctness:** 3
**Technical Novelty And Significance:** 2
**Empirical Novelty And Significance:** 2
**Recommendation:** 6
**Confidence:** 2

**Main Review:**

Pros:
- The proposed method is generic and easily incorporated to existing methods in a plug-and-play manner.
- Empirical results on HICO and GQA show consistent improvements in performance over baselines. The proposed method also achieves some good results in systematic generalization settings.


Cons:

(1) Proposed method
- The scientific novelty of this paper seems limited. The overall structure of the proposed model is almost similar to the "self-distillation with no labels" approach (Caron et al.,2021) and EsViT (Li et al.,2021). The contribution is mostly about the use of their concept-feature dictionary as a visual knowledge base to support retrieving information on the fly. The use of the dictionary is more similar to the concept of knowledge base used by [1] rather than memory bank mechanisms as the proposed method only reads out information from the dictionary without refining the information stored in the dictionary.
- The authors should explain more how they pick the suitable image feature f from a set of a queue Q_i. It is an important step in the proposed model, and the description for their two sampling tactics is not detailed enough.
- Although the reported numbers look great, it is hard to explain why the visual concepts given by ViT are good for visual relation reasoning. ViT does not actually rely on actual “visual objects”, for example, RoI poolings in Faster R-CNN. I think it is more about the more powerful representation learning pretrained on a large number of images which benefits all tasks regardless.

(2) Experiments
- My main concern about experimental results is that the improvement of the proposed model may come from the Visual Transformer backbone. It would be fair if the authors also evaluate their concept-guided approach with the ResNet-101 backbone or actual visual objects given by Faster R-CNN.
- Even using the state-of-the-art ViT backbone, the performance of the proposed model is still worse than state-of-the-art methods on the GQA dataset in both original and systematic settings, and only has sightly improvement on the HICO dataset in the original setting.
- CLOSURE probably is a better benchmark for evaluating systematic generalization.

[1] Hudson DA, Manning CD. Compositional attention networks for machine reasoning. ICLR 2019.

[2] Bahdanau, Dzmitry, et al. "Closure: Assessing systematic generalization of clevr models." arXiv preprint arXiv:1912.05783 (2019).

**Summary Of The Paper:**

The paper proposes a new concept-guided approach for visual relational reasoning. Particularly, it uses a concept-feature dictionary to store and retrieve the feature of an image by the corresponding visual concepts. Then, the concept-guided features are integrated into the training process with the global and local auxiliary tasks. Experimental results on HICO and GQA in both original and systematic settings evidenced the effectiveness of the proposed method against existing works.

**Summary Of The Review:**

The paper appears to be limited in novelty and contribution.

---

### Decision · Program_Chairs · 2022-01-20

**Decision:**

Accept (Poster)

**Comment:**

Three experts reviewed the paper and all recommended acceptance. Based on the reviewers' feedback, the decision is to recommend the paper for acceptance. However, the reviewers did raise some valuable concerns that should be addressed in the final camera-ready version of the paper. For example, the discussion about or comparison with related works, clarity of the writing, suggested experiments, etc. The authors are encouraged to make the necessary changes to the best of their ability. We congratulate the authors on the acceptance of their paper!